# SNP rs2920280 in *PSCA* Is Associated with Susceptibility to Gastric Mucosal Atrophy and Is a Promising Biomarker in Japanese Individuals with *Helicobacter pylori* Infection

**DOI:** 10.3390/diagnostics12081988

**Published:** 2022-08-16

**Authors:** Hajime Isomoto, Takuki Sakaguchi, Tatsuo Inamine, Shintaro Takeshita, Daisuke Fukuda, Ken Ohnita, Tsutomu Kanda, Kayoko Matsushima, Tetsuro Honda, Takaaki Sugihara, Tatsuro Hirayama, Kazuhiko Nakao, Kazuhiro Tsukamoto

**Affiliations:** 1Department of Gastroenterology and Nephrology, Faculty of Medicine, Tottori University, 36-1 Nishi-cho, Yonago 683-8504, Japan; 2Department of Gastroenterology and Hepatology, Nagasaki University Graduate School of Biological Sciences, 1-7-1 Sakamoto, Nagasaki 852-8501, Japan; 3Department of Pharmacotherapeutics, Nagasaki University Graduate School of Biomedical Sciences, 1-7-1 Sakamoto, Nagasaki 852-8501, Japan; 4Department of Surgical Oncology, Nagasaki University Graduate School of Biological Sciences, 1-7-1 Sakamoto, Nagasaki 852-8501, Japan; 5Fukuda Yutaka Clinic, 3-5 Hamaguchi-machi, Nagasaki 852-8107, Japan; 6Shunkaikai Inoue Hospital, 6-12 Takara-machi, Nagasaki 850-0045, Japan; 7Department of Gastroenterology, Nagasaki Medical Harbor Center, 6-39 Shinchi-machi, Nagasaki 850-8555, Japan

**Keywords:** prostate stem cell antigen, single-nucleotide polymorphisms, gastric mucosal atrophy, *Helicobacter pylori* infection, association study, gastric carcinogenesis, pepsinogen, Kimura–Takemoto classification, biomarker, gastric cancer screening

## Abstract

*Helicobacter pylori* infection results in gastric cancer (GC) with gastric mucosal atrophy (GMA). Some single-nucleotide polymorphisms (SNPs) in the prostate stem cell antigen gene (*PSCA*) are associated with GC and duodenal ulcers. However, the relationship of other identified SNPs in *PSCA* with these diseases remains unclear. Herein, the association between *PSCA* SNPs and GMA among 195 Japanese individuals with *H. pylori* infection was evaluated. The definition of GMA or non-GMA was based on serum pepsinogen levels or endoscopic findings. Five tag *PSCA* SNPs were analyzed using PCR high-resolution melting curve analysis with nonlabelled probes. The frequencies of alleles and the genotypes of each tag SNP were compared between the GMA and non-GMA groups. Subsequently, a genetic test was performed using associated SNPs as biomarkers to detect patients developing GMA. Two tag *PSCA* SNPs (rs2920280 and rs2294008) were related to GMA susceptibility. Individuals with the rs2920280 G/G genotype or the rs2294008 T/T genotype in *PSCA* had 3.5- and 2.1-fold susceptibility to GMA, respectively. In conclusion, SNP rs2920280 is a possible biomarker for detecting individuals developing GMA. *PSCA* polymorphisms may be useful biomarkers for predicting GMA linked to GC risk and a screening endoscopy strategy to detect GC related to early stage *H. pylori* associated GMA.

## 1. Introduction

*Helicobacter pylori* colonizes the gastric mucosa and infects no less than half of the entire human population [1]. Persistent *H. pylori* infection causes chronic stomach inflammation, eventually leading to gastric mucosal atrophy (GMA) [2]. Gastric cancer develops through a multistep process known as the persistent gastritis–GMA carcinogenic pathway, which is triggered by *H. pylori* infection [3]. Some individuals infected with this bacterium develop gastric cancer with advanced GMA or a precursor toward gastric cancer in a particular intestinal type [3,4]. This may be due to virulence differences among the *H. pylori* strains; environmental factors, such as salt intake and smoking; and the host’s genetic predisposition, including various single-nucleotide morphisms (SNPs) [5].

Prostate stem cell antigen (*PSCA*) is a membrane protein binding to cell surfaces through a glycosylphosphatidylinositol anchor [6]. *PSCA* expression is upregulated in prostate, bladder, and pancreatic cancer, but is downregulated in esophageal and gastric cancer [7,8,9,10,11,12,13,14]. *PSCA* knockdown in bladder cancer suppresses cancer cell proliferation and induces the expression of anti-inflammatory cytokines, such as interferon α/β (oncogenic effect) [15]. However, *PSCA* has growth inhibition activity on gastric cancer cells. This suggests that the physiological functions of *PSCA* include both oncogenic and tumor-suppressor effects in tissue- and cell-specific manners [6,7,8,9,10,11,12,13,14,15,16].

Genome-wide studies have revealed the association of SNP rs2294008 in *PSCA* with representative *H. pylori* related diseases, including gastric cancer and duodenal ulcer [7,8]. Although most studies on the relationship between *PSCA* genetic predisposition and gastroduodenal diseases and gastric cancers have focused on rs2294008 in *PSCA*, there is information on the association between the risk of gastric cancer and other *PSCA* genetic variations in the literature [17]. In the present work, an association study was conducted between several SNPs in *PSCA* and GMA susceptibility in Japanese individuals with *H. pylori* infection to identify new host genetic factors. The associated SNPs in *PSCA* can be used as new genetic biomarkers for predicting GMA development, which is a precancer lesion. In this study, we aimed to detect the useful stratification marker to assess gastric cancer risk related to *H. pylori* associated GMA.

## 2. Materials and Methods

### 2.1. Study Participants

We enrolled 195 individuals diagnosed as *H. pylori* positive with a serum *H. pylori antibody* titer ≥ 10 U/mL (E-plate Eiken *H. pylori* antibody II; Eiken Chemical, Tokyo, Japan) among 502 individuals who underwent upper gastrointestinal endoscopy for a healthy checkup between August and December 2013. The study requirements stated that participants must be below 80 years old without a history of stomach surgery or *H. pylori* eradication therapy. The study included 109 women and 86 men, with a mean age of 56.9 (29–77) years.

### 2.2. GMA Classification

GMA-positive individuals were determined using two methods: the serological pepsinogen (PG) method and the Kimura–Takemoto endoscopic classification system [18,19]. Participants with serum PG I levels ≤ 70 μg/L and a serum PG I/II ratio ≤ 3.0 were classified into the GMA group, while those who did not meet any of these requirements were classified into the non-GMA group [18].

Participants who were diagnosed through the endoscopic findings of C3 or O1 to O3 according to the Kimura–Takemoto endoscopic classification were classified into the GMA group, and those who were diagnosed through those of C1 or C2 were classified into the non-GMA group [19].

### 2.3. Selection of Tag SNPs in PSCA

*PSCA* (OMIM #602470, located at 8q24.3) was selected as the candidate gene for GMA susceptibility. Five tag SNPs (rs6471587, rs2920280, rs2294008, rs2976391, and rs3736001) were genotyped based on information concerning SNPs in *PSCA* according to previous methods [20,21]. In brief, we extracted the SNPs reported in Japanese people according to the International HapMap database in the region from the candidate gene and promoter to 2 kb upstream. Then, we extracted five SNPs with minor allele frequency over 0.05 and that met linkage disequilibrium (r2 > 0.8) using Haploview ver. 4.2 as tag SNPs. The gene structure and locations of the tag SNP sites in *PSCA* are shown in Figure 1.

### 2.4. SNP Genotyping

Each participant’s peripheral blood was stored in a blood collection tube with EDTA. Genomic DNA was extracted from the peripheral blood of each participant using Nucleo Spin^®^ (Takara, Shiga, Japan) according to the manufacturer’s instructions. Genotyping of the five tag SNPs in *PSCA* was carried out by the probe-based high-resolution melting method [22,23] using the primers, probes, and annealing temperatures listed in Table 1.

### 2.5. Statistical Analyses

The mean participant age, presented as the mean ± standard deviation, was compared between the GMA and non-GMA groups using the Mann–Whitney *U* test. Likewise, sex was compared using the chi-square test. Statistical analyses were performed using SPSS 17 (SPSS Japan, Tokyo, Japan) or Prism 6 (GraphPad Software Inc., La Jolla, CA, USA).

A chi-square test was performed based on the expectation-maximization algorithm using the SNPAlyze^®^ ver. 7.1 standard software package (DynaCom Inc., Yokohama, Japan) to determine whether each tag SNP was in Hardy–Weinberg equilibrium. The frequencies of alleles and genotypes of each tag SNP between the GMA and non-GMA groups were compared by chi-square or Fisher’s exact test with an odds ratio (OR) and 95% confidence interval in three different inheritance models: the minor allele, minor allele dominant, and minor allele recessive using the SNPAlyze^®^ ver.7.1 standard software package and SPSS 17.

In addition to univariate analyses, multivariate logistic regression analysis was conducted to assess the interaction of genetic and environmental factors with GMA susceptibility using SPSS 17. *p* < 0.05 was considered to indicate significance.

## 3. Results

### 3.1. Association between PSCA Polymorphisms and GMA Susceptibility by the PG Method

The frequencies of alleles and genotypes of the five tag SNPs in *PSCA* showed no significant differences in the allele and genotype frequencies between the GMA (*n* = 92) and non-GMA (*n* = 103) groups in the three inheritance models: allele, dominant, and recessive models (Table 2).

### 3.2. Association between PSCA Polymorphisms and GMA Susceptibility Based on Endoscopic Findings According to the Kimura–Takemoto Classification

The frequency of the minor G allele of rs2920280 in *PSCA* in the minor allele model was significantly higher in the GMA group (*n* = 123, 50.8%) than in the non-GMA group (*n* = 72, 37.5%, *p* = 0.011, OR = 1.722; Table 3), thereby indicating a 1.7-fold increase in susceptibility to GMA. Likewise, the frequency of the minor homozygosity (G/G genotype of rs2920280 in *PSCA*) in the minor allele recessive model was significantly higher in the GMA group than in the non-GMA group (27.7% vs. 9.7%, *p* = 0.003, OR = 3.547; Table 3), showing that there was an approximately 3.5-fold increase in susceptibility to GMA. In addition, the frequency of the minor C allele of rs2294008 in *PSCA* in the minor allele model was significantly lower in the GMA group than in the non-GMA group (35.8% vs. 46.5%, *p* = 0.036, OR = 0.640; Table 3), indicating a 0.64-fold susceptibility to GMA. Thus, the presence of the C allele of rs2294008 in *PSCA* indicated an approximately 1.6-fold increased resistance to GMA development. Moreover, the frequency of the T/C or C/C genotype of rs2294008 in *PSCA* in the minor allele dominant model was significantly lower in the GMA group than in the non-GMA group (61.0% vs. 76.4%, *p* = 0.028, OR = 0.483; Table 3), indicating a 0.48-fold susceptibility to GMA. Thus, the possession of the T/C or C/C genotype of rs2294008 in *PSCA* indicated an approximately 2.1-fold increased resistance to GMA development. Conversely, the possession of the major homozygous T/T genotype of rs2294008 in *PSCA* indicated an approximately 2.1-fold increase in susceptibility to GMA.

### 3.3. Comparison of Clinical Characteristics between GMA and Non-GMA Groups Based on Endoscopic Findings According to Kimura–Takemoto Classification

According to the mean age, patients in the GMA group were approximately six years older than those in the non-GMA group (59.1 vs. 53.2, *p* = 0.0002; Table 3), indicating that the older participants showed greater GMA susceptibility. However, there was no significant difference in sex between the GMA and non-GMA groups (Table 3).

### 3.4. Contribution of Gene–Environment Interactions to GMA Susceptibility

The results of univariate analyses revealed that the G/G genotype of rs2920280 in *PSCA*, the T/T genotype of rs2294008 in *PSCA*, and age contributed to GMA susceptibility (Table 3). The results of ultivariate analysis indicated that three independent factors contributed to susceptibility to GMA with statistical significance: the G/G genotype of rs2920280 in *PSCA*, T/T genotype of rs2294008 in *PSCA*, and age (Table 4 and Table 5). Furthermore, genetic testing indicated that the G/G genotype of rs2920280 in *PSCA,* with a sensitivity of 27.6%, specificity of 90.3%, positive predictive value (PPV) of 82.9%, and negative predictive value of 42.2%, was more useful as a biomarker than the T/T genotype of rs2294008 in *PSCA* (Table 6).

## 4. Discussion

The present study suggests that *PSCA* polymorphisms contributed to the onset and/or development of GMA in *H. pylori* positive Japanese individuals who were endoscopically evaluated using the Kimura–Takemoto classification. Individuals with the G/G genotype of rs2920280 or the T/T genotype of rs2294008 in *PSCA* showed a 3.5- or 2.1-fold increase in susceptibility to GMA, respectively. The results of multivariate analysis also indicated that age is an independent factor that significantly contributed to susceptibility to GMA. This result agrees with that of a previous study, in which over 30% of patients’ gastric mucosa worsened during the atrophic and intestinal metaplasia phases with increasing age [24].

*PSCA* is upregulated in bladder cancer, renal carcinoma, and pancreatic cancer, and is downregulated in gastric, esophagus, and gallbladder cancer [25,26]. In the field of gastric and esophagus cancer, the downregulation of *PSCA* promotes the proliferation of cancer cells, and patients whose *PSCA* is downregulated have poor prognoses [26,27]. These data suggest that *PSCA* has a tumor-suppressing effect in gastric and esophagus cancer development. Zhang et al. reported that *PSCA* arrests cell cycle progression and promotes cell differentiation in esophageal squamous cell carcinoma by binding to retinoblastoma 1-inducible coiled-coil 1 (RB1CC1) [26]. Although the precise function of *PSCA* in gastric cancer remains unknown [28], the fact that *PSCA* protein is mainly expressed in the isthmus, which contains stem cells and precursors of cells, and in a variety of cell lineage and differentiation stage, indicates that *PSCA* is involved in the differentiation [7]. Moreover, a slower growth rate of the *PSCA* expression cells than that without *PSCA* expression suggests that *PSCA* arrests the cell cycle without cell death [7].

rs2294008T reduces *PSCA* mRNA expression rather than rs2294008C in bladder cancer [29]. The T allele of rs2294008, which is considered a risk allele in gastric cancer, upregulated the expression of *PSCA* mRNA compared with the C allele of rs22904008 both in normal gastric and gastric cancer tissues, and the prognosis worsened with increased mRNA expression [30]. These contradictory results may be due to the qualitative change in traits in *PSCA* [30]. rs2294008T produces a long *PSCA*, which is nine amino acids longer than the short *PSCA* produced by rs2294008C [8]. Long *PSCA* is located on the membrane and has a growth-promoting ability, whereas short *PSCA* is in the cytoplasm and easily disassembles [8]. The signal analysis of long *PSCA* is not fully elucidated. Further investigation is required.

SNP rs2920280 may be more useful as a prognostic biomarker for detecting individuals at risk of developing GMA among *H. pylori* infected individuals than SNP rs2294008. The G/G genotype of rs2920280 in *PSCA* showed a higher OR, lower *p*-value, higher specificity, and higher PPV than the T/T genotype of rs2294008, indicating that the G/G genotype is a more useful biomarker despite its lower sensitivity (Table 6).

The lower sensitivity of the biomarker rs2294008 implies the detection of only one-quarter of individuals developing GMA. Nevertheless, the higher PPV implies that approximately 83% of *H. pylori* positive individuals with the G/G genotype of rs2920280 in *PSCA* develop GMA. These high-risk individuals may require careful observation and/or occasional treatment for developing GMA and gastric cancer because early initiation of eradication treatment may delay the progression toward advanced GMA [31,32,33].

SNP rs2920280 is located within intron 1 of *PSCA* (Figure 1). The results of GWAS analysis revealed that nine *PSCA* SNPs, including rs2294008 and rs2920283, are identified as risk SNPs for gastric atrophy and severe gastric atrophy [34]. Sophie K.M. Heinrichs et al. conducted eQTL analysis using 143 subjects of German descent with intestinal metaplasia [35]. They reported that the lead eQTL for *PSCA* is rs2920283, and an upregulated *PSCA* mRNA expression in gastric tissue was observed in gastric cancer risk allele carriers. Although we do not have data indicating the frequency of the genotype at rs2920283, rs2920280, which has linkage distribution (D’ = 0.971, r2 = 0.600) to rs2920283 according to Ensembl (https://asia.ensembl.org/Homo_sapiens/Info/Index accessed on 29 July 2022), may increase *PSCA* mRNA. However, the localization and function of *PSCA* with rs2920280G are not fully elucidated. Further experiments are needed.

Although early detection and eradication of *H. pylori* infection in adolescents is important, the benefit and harm of the eradication therapy should be always considered. In addition, these two SNP biomarkers would be useful if combined with the ABC methods, which are used to evaluate the conditions of the stomach based on the noninvasive methods of measuring pepsinogen and *H. pylori* antibody [36]. Stratifying the patients between the high- and the low-risk groups would be helpful for deciding whether the patients should take medicine or encourage frequent endoscopic surveillance for gastric cancer identification.

Our study had some limitations. First, the sample size was quite small. The minor allele frequencies of associated SNPs ranged from 10% to 20%, the OR was 2 to 3.5, and the *p*-values ranged from 0.03 to 0.003 in this study. Further prospective studies with larger sample sizes and higher statistical power are warranted to confirm the significant positive association between *PSCA* SNPs and the development to GMA. Second, the *PSCA* SNPs that met linkage disequilibrium (r2 > 0.8) were selected as tag SNPs. We could not deny that rs2294008 and rs2920280 were completely in linkage disequilibrium because these two SNPs were located within a short distance from each other. We separately included these two SNPs in the analysis in association with environmental factors and age. Third, this study did not include a correlation analysis between the role of *PSCA* genetic variations in GMA and gastric cancer, and there was no follow-up survey of the enrolled subjects. In addition, the CagA virulence factor of the bacterium was not examined, although almost all Japanese people harbor the East-Asia-type virulent *H. pylori* strain [37]. Nevertheless, such genetic testing may provide future directions for precision medicine for individuals more likely to develop GMA and encourage frequent endoscopic surveillance for gastric cancer identification or its prevention by eradication therapy. Again, adolescents with risky SNPs of *PSCA* would potentially be at risk of GMA development, linking to GC decades later.

## 5. Conclusions

Our study findings indicate that *PSCA* SNPs rs2920280 might be associated with GMA in Japanese individuals with *H. pylori* infection. *PSCA* SNPs consisting of the well-known T/T genotype of rs2294008 and the newly identified G/G genotype of rs2920280 can be a useful biomarker for predicting GMA development. Accordingly, we encourage *H*. *pylori* infected individuals, who cannot or did not eradicate *H. pylori* and possess these two SNPs, to undergo timely endoscopic screening for early-stage gastric cancer. Further investigations regarding *PSCA* polymorphisms and their functions are required to provide pathophysiological findings and establish reliable biomarkers, finally leading to the development of new therapeutic GMA drugs to reduce gastric cancer risk.

## Figures and Tables

**Figure 1 diagnostics-12-01988-f001:**
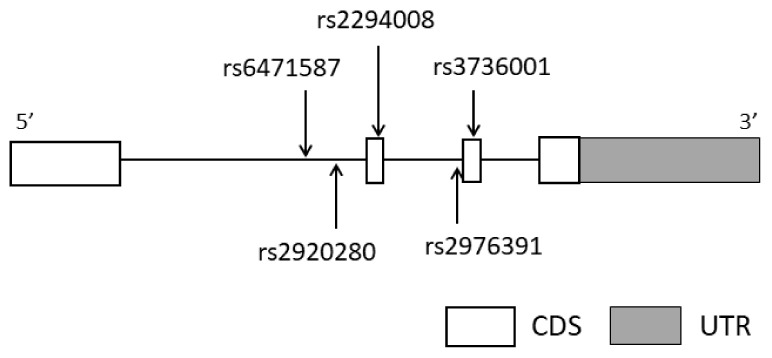
Gene structure and locations of the genotyped tag single-nucleotide polymorphism (SNP) sites in prostate stem cell antigen gene (*PSCA*). The horizontal bar indicates the genomic sequence of *PSCA.* Open boxes and lines represent exons and introns, respectively, with the widths not representative of the base-pair length. The dark gray box shows the untranslated region (UTR). Vertical arrows indicate genotyped tag SNP sites in this study.

**Table 1 diagnostics-12-01988-t001:** Information regarding genotyping of tag SNPs in *PSCA*.

SNP	Primer Sequences (5′ to 3′)		Annealing Temperature (°C)
	forward	TCAGTGCACAGCTTGATGGT	
rs6471587	reverse	CCCCGTATAACCCAGACATT	56
	probe	CTGGTTGGCAATCGCTTGCAAGAGTTATC	
	forward	GGAACAGCCAGCGTTAACAT	
rs2920280	reverse	CATCTTCTGGTTGGCAATCG	55
	probe	GTCTCCACAACCCGGTATAACCCAGA	
	forward	CCTCACTGGCTCCAGGAAAC	
rs2294008	reverse	AAGCCTGCCATCAACAGGG	58
	probe	CCACCAGTGACCATGAAGGCTGTGCT	
	forward	GGAGGAGACATTGAGGGTGA	
rs2976391	reverse	TTTGCAGGAGTAGCACAGCA	57
	probe	CAGCCTCTGAGGCCCCTCCCACTCCA	
	forward	TGCTGTGCTACTCCTGCAAA	
rs3736001	reverse	AGGTGGAAGGAAAACAGCAC	56
	probe	TGCCTGCAGGTGGAGAACTGCACCC	

Abbreviations: SNP, single-nucleotide polymorphism; *PSCA*, prostate stem cell antigen.

**Table 2 diagnostics-12-01988-t002:** Allele and genotype frequencies of tag SNPs in *PSCA* for the GMA and non-GMA groups at the selected serum PG I level and the specified PG I/II ratio.

SNP	Genotype	GMA (%)	Non-GMA (%)	Inheritance Model	OR (95% CI)	*p*-Value
		PG I ≤ 70 and PG I/II ≤ 3	Others			
Clinical characteristics	Number	92	103			
	Age, mean ± SD	59.0 ± 9.6	55.0 ± 11.0			0.0055
	Male/female	37/55	49/54			0.302
rs6471587	G allele *	24 (13.0)	32 (15.5)	Allele	0.816 (0.456–1.467)	0.484
	C/C	70 (76.1)	74 (71.9)			
	C/G	20 (21.7)	26 (25.2)	Dominant	0.802 (0.432–1.520)	0.501
	G/G	2 (2.2)	3 (2.9)	Recessive	0.741 (0.129–3.700)	1.000
rs2920280	G allele *	92 (50.0)	87 (42.2)	Allele	1.368 (0.922–2.037)	0.124
	C/C	24 (26.1)	33 (32.0)			
	C/G	44 (47.8)	53 (51.5)	Dominant	1.336 (0.732–2.502)	0.362
	G/G	24 (26.1)	17 (16.5)	Recessive	1.785 (0.900–3.481)	0.101
rs2294008	C allele *	69 (37.5)	86 (41.7)	Allele	0.837 (0.553–1.259)	0.392
	T/T	34 (37.0)	31 (30.1)			
	T/C	47 (51.1)	58 (56.3)	Dominant	0.735 (0.398–1.342)	0.310
	C/C	11 (11.9)	14 (13.6)	Recessive	0.863 (0.3588–1.997)	0.733
rs2976391	A allele *	33 (17.9)	39 (18.9)	Allele	0.936 (0.551–1.563)	0.800
	C/C	61 (65.2)	67 (65.1)			
	C/A	29 (32.6)	33 (32.0)	Dominant	0.946 (0.517–1.711)	0.854
	A/A	2 (2.2)	3 (2.9)	Recessive	0.7461 (0.129–3.700)	1.000
rs3736001	A allele *	15 (8.2)	24 (11.7)	Allele	0.673 (0.350–1.346)	0.250
	G/G	77 (83.7)	80 (77.7)			
	G/A	15 (16.3)	22 (21.3)	Dominant	0.678 (0.326–1.400)	0.289
	A/A	0 (0)	1 (1.0)	Recessive	0.000 (0.099–10.080)	1.000

Asterisks indicate the number of minor alleles of each SNP but not genotypes. Abbreviations: SNP, single-nucleotide polymorphism; *PSCA*, prostate stem cell antigen; GMA, gastric mucosal atrophy; PG, pepsinogen; OR, odds ratio; CI, confidence interval; SD, standard deviation.

**Table 3 diagnostics-12-01988-t003:** Frequencies of alleles and genotypes of tag SNPs in *PSCA* for the GMA and non-GMA groups under the Kimura–Takemoto classification.

SNP	Genotype	GMA (%)	Non-GMA (%)	Inheritance Model	OR (95% CI)	*p*-Value
		C3 or O1-3	C1 or C2			
Clinical characteristics	Number	123	72			
	Age, mean ± SD	59.1 ± 9.1	53.2 ± 11.6			0.0002
	Male/female	55/68	31/41			0.822
rs6471587	G allele *	33 (13.4)	23 (16.0)	Allele	0.815 (0.466–1.485)	0.487
	C/C	93 (75.6)	51 (70.8)			
	C/G	27 (22.0)	19 (26.4)	Dominant	0.783 (0.406–1.480)	0.464
	G/G	3 (2.4)	2 (2.8)	Recessive	0.875 (0.175–5.028)	1.000
rs2920280	G allele *	125 (50.8)	54 (37.5)	Allele	1.722 (1.122–2.629)	0.011
	C/C	32 (26.0)	25 (34.7)			
	C/G	57 (46.3)	40 (55.6)	Dominant	1.513 (0.801–2.808)	0.197
	G/G	34 (27.7)	7 (9.7)	Recessive	3.547 (1.545–9.006)	0.003
rs2294008	C allele *	88 (35.8)	67 (46.5)	Allele	0.640 (0.424–0.966)	0.036
	T/T	48 (39.0)	17 (23.6)			
	T/C	62 (50.4)	43 (59.7)	Dominant	0.483 (0.246–0.905)	0.028
	C/C	13 (10.6)	12 (16.7)	Recessive	0.591 (0.249–1.375)	0.219
rs2976391	A allele *	39 (15.9)	33 (22.9)	Allele	0.634 (0.376–1.046)	0.083
	C/C	86 (69.9)	42 (58.3)			
	C/A	35 (28.5)	27 (37.5)	Dominant	0.602 (0.322–1.131)	0.100
	A/A	2 (1.6)	3 (4.2)	Recessive	0.380 (0.067–1.909)	0.360
rs3736001	A allele *	22 (8.9)	17 (11.8)	Allele	0.734 (0.388–1.4361)	0.363
	G/G	102 (82.9)	55 (76.4)			
	G/A	20 (16.3)	17 (23.6)	Dominant	0.666 (0.318–1.344)	0.266
	A/A	1 (0.8)	0 (0)	Recessive	0.000 (0.065–15.380)	1.000

Asterisks indicate the number of minor alleles of each SNP but not genotypes. Abbreviations: SNP, single-nucleotide polymorphism; *PSCA*, prostate stem cell antigen; GMA, gastric mucosal atrophy; PG, pepsinogen; OR, odds ratio; CI, confidence interval; SD, standard deviation.

**Table 4 diagnostics-12-01988-t004:** Contribution of gene–environment interaction to GMA with regard to rs2920280 in *PSCA*.

Factor	Factor Comparison	
	OR (95% CI)	*p*-Value
Age	1.060 (1.028–1.093)	<0.001
G/G genotype of rs2920280 in *PSCA*	3.665 (1.499–8.911)	0.004

Abbreviations: GMA, gastric mucosal atrophy; *PSCA*, prostate stem cell antigen; OR, odds ratio; CI, confidence interval.

**Table 5 diagnostics-12-01988-t005:** Contribution of gene–environment interaction to GMA with regard to rs2294008 in *PSCA*.

Factor	Factor Comparison	
	OR (95% CI)	*p*-Value
Age	1.058 (1.026–1.091)	<0.001
T/T genotype of rs2294008 in *PSCA*	1.992 (1.016–3.906)	0.045

Abbreviations: GMA, gastric mucosal atrophy; *PSCA*, prostate stem cell antigen; OR, odds ratio; CI, confidence interval.

**Table 6 diagnostics-12-01988-t006:** Evaluation of a genetic test using the associated tag SNPs as a biomarker for susceptibility to GMA.

Biomarker	OR (95% CI)	*p*-Value	Sensitivity (%)	Specificity (%)	PPV (%)	NPV (%)
G/G genotype of rs2920280	3.547 (1.545–9.006)	0.003	27.6	90.3	82.9	42.2
T/T genotype of rs2294008	2.070 (1.105–4.065)	0.028	39	76.4	73.8	42.3

Abbreviations: SNPs, single-nucleotide polymorphisms; GMA, gastric mucosal atrophy; *PSCA*, prostate stem cell antigen; OR, odds ratio; CI, confidence interval; PPV, positive predictive value; NPV, negative predictive value.

## Data Availability

Not applicable.

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
