# Peer review of "SNP rs2920280 in PSCA Is Associated with Susceptibility to Gastric Mucosal Atrophy and Is a Promising Biomarker in Japanese Individuals with Helicobacter pylori Infection"

_diagnostics, 2022, doi:10.3390/diagnostics12081988_

Round 1
Reviewer 1 Report
A paper entitled "SNP rs2920280 in PSCA is associated with susceptibility to gastric mucosal atrophy and is a promising biomarker in Japanese individuals with Helicobacter pylori infection" by Takuki Sakaguchi et al. It was demonstrated that it is a potential GMA susceptibility gene in Japanese with H. pylori. An SNP consisting of the well-known T / T genotype of rs2294008 and the newly identified G / G genotype of rs2920280 can be used as a biomarker for predicting GMA development.
Although this study is of some value, the findings of this study appear to be a lot of preliminary data and have not been adequately discussed as currently available biomarkers for PSCA polymorphisms. As a result, the authors have shown limited interest.
Therefore, there are areas that need improvement.
Main comments
1. Usually, two methods, serological pepsinogen (PG) method and H. pylori antibody test, can be used to predict GMA-positive individuals. It is not clear why the PSCA SNP, which consists of the well-known T / T genotype of rs2294008 and the newly identified G / G genotype of rs2920280, is superior to these existing trials.
2. Identification of new biomarkers for early detection of gastric cancer is expected. In this study, the advantages of PSCA SNPs analysis over conventional methods for detecting gastric cancer are not fully mentioned.
3. Early detection of H. pylori infection in GMA-negative adolescents is important in the real world. What about PSCA's SNPs analysis in this regard?
Minor comments
1. The description of some limitations in this study (lines 225-239) is somewhat verbose and needs to be shortened.
Author Response
- Usually, two methods, serological pepsinogen (PG) method and H. pylori antibody test, can be used to predict GMA-positive individuals. It is not clear why the PSCA SNP, which consists of the well-known T/T genotype of rs2294008 and the newly identified G/G genotype of rs2920280, is superior to these existing trials.
Thank you for your advice. However, In this study, we aimed to detect the other potentially useful stratification marker candidate to assess gastric cancer risk related to H. pylori-associated GMA than pre-existing serological tests. Again, circulating PG measurement remains uncovered by the Japanese health insurance. Also, serological anti-HP antibody titer measurement has a remaining problem as for threshold for negativity. To undergo anti-HP eradication, the Japanese patient have to undergo gastroscopy under the health insurance. Among never-infected, current HP-infected and its eradicated individuals, the 2 existing serological tests would not be predictable for GMA.
- Identification of new biomarkers for early detection of gastric cancer is expected. In this study, the advantages of PSCA SNPs analysis over conventional methods for detecting gastric cancer are not fully mentioned.
Thank you for your comments. In this study, we could not follow up with the participants. Therefore, we aimed to detect the biomarkers for predicting GMA development, which is a precancer lesion. An advantage of SNPs such as PSCA can be that SNPs are stable during lifelong that not be affected by eradications including via the anti-bacterial regimen or naturally.
- Early detection of H. pylori infection in GMA-negative adolescents is important in the real world. What about PSCA's SNPs analysis in this regard?
Although early detection and eradication of H. Pylori infection in adolescents is important, the benefit and harm of the eradication therapy should be always considered. And more, these two SNPs biomarkers would be useful if combined with the ABC methods, which are used to evaluate conditions of the stomach based on the noninvasive methods of measuring pepsinogen and H. Pylori antibody. Stratifying the patients between the high-risk group and the low-risk group would be helpful for deciding whether the patients should take medicine or encourage frequent endoscopic surveillance for gastric cancer identification. We appreciate the reviewer’s insightful comment to use PSCA’s SNP in adolescents in whom GMA would not be so advanced at that time point. We add ‘Adolescents with risky SNPs of PSCA would have potential risk of GMA development linking to GC decades later’.
Minor comments
- The description of some limitations in this study (lines 225-239) is somewhat verbose and needs to be shortened.
Thank you for your advice. I had changed the limitation to be shortened.
Reviewer 2 Report
Manuscript ID: diagnostics-1823431
This short manuscript by Takuki Sakaguchi et al. report that SNP rs2920280 is a possible biomarker for detecting individuals developing gastric mucosal atrophy. The manuscript is well-written and the findings described are of interest. However, several points listed below should be reconsidered for publication.
(1) Sample size is too small to conclude. Well-designed functional studies are required to confirm the findings.
(2) Potential mechanisms should be stated.
(3) Important papers (PMID: 30753327, PMID: 29892961, PMID: 29028942, and PMID: 30191681) are not listed. These should be cited and discussed.
Author Response
(1) Sample size is too small to conclude. Well-designed functional studies are required to confirm the findings.
I recognize that the sample size is one of the limitations of this study. In this study, we could only demonstrate the “PSCA SNPs rs2920280 might be potentially associated with GMA in Japanese individuals with H. pylori infection.” As we mentioned perspective, it is warrant be validated in the prospective well-designed studies; we are to entertain in the multi-center prospective study.
(2) Potential mechanisms should be stated.
Thank you for your advice. I changed my discussion as follows.
“Li-Yi Zhang et al. reported that PSCA arrests cell cycle progression and promotes cell differentiation in esophageal squamous cell carcinoma by binding to retinoblastoma 1-inducible coiled-coil 1 (RB1CC1). Although the precise function of PSCA in gastric cancer remains unknown, the fact that PSCA protein is mainly expressed in the isthmus, which contains stem cells and precursors of cells, and in a variety of cell lineage and differentiation stage indicate PSCA would involve in the differentiation. And a slower growth rate of the PSCA expression cells than on without PSCA expression suggests that PSCA would arrest the cell cycle without cell death. The rs2294008T reduces PSCA mRNA expression rather than rs2294008C in bladder cancer. On the other hand, the T allele of rs2294008, which is considered a risk allele in gastric cancer, upregulated the expression of PSCA mRNA compared with the C allele of rs22904008 both in normal gastric and gastric cancer tissue, and the prognosis gets worse with increased mRNA expression. These contradictory results may be due to the qualitative traits change in PSCA. The rs2294008T produces a long PSCA, which has nine amino acids longer than the short PSCA produced by the rs2294008C. The long PSCA is located on the membrane and has a growth-promoting ability, whereas the short PSCA is in the cytoplasm and is easy to be disassembled. The signal analysis of long PSCA is not fully elucidated. Further investigation is required.” “GWAS analysis reveals that nine PSCA SNPs, including rs2294008 and rs2920283, are identified as risk SNPs for gastric atrophy (GA) and severe GA [34]. Sophie K.M. Hein-richs et al. had conducted the eQTL analysis using 143 subjects of German descent with intestinal metaplasia [35]. They reported that the lead eQTL for PSCA is rs2920283, and an upregulated PSCA mRNA expression in gastric tissue was observed in gastric cancer risk allele carriers. Although we don’t have data indicating the frequency of genotype at rs2920283, rs2920280, which has linkage distribution (D’=0.971, r2= 0.600) to the rs2920283 according to the Ensembl (https://asia.ensembl.org/Homo_sapiens/Info/Index), may in-crease PSCA mRNA. However, the localization and function of PSCA with the rs2920280G are not fully elucidated. Further experiments are needed.”
(3) Important papers (PMID: 30753327, PMID: 29892961, PMID: 29028942, and PMID: 30191681) are not listed. These should be cited and discussed.
Thank you for your advice. I cite these papers indicated above as references.
I changed my discussion as follows.
“Li-Yi Zhang et al. reported that PSCA arrests cell cycle progression and promotes cell differentiation in esophageal squamous cell carcinoma by binding to retinoblastoma 1-inducible coiled-coil 1 (RB1CC1). Although the precise function of PSCA in gastric cancer remains unknown, the fact that PSCA protein is mainly expressed in the isthmus, which contains stem cells and precursors of cells, and in a variety of cell lineage and differentiation stage indicate PSCA would involve in the differentiation. And a slower growth rate of the PSCA expression cells than on without PSCA expression suggests that PSCA would arrest the cell cycle without cell death.
The rs2294008T reduces PSCA mRNA expression rather than rs2294008C in bladder cancer. On the other hand, the T allele of rs2294008, which is considered a risk allele in gastric cancer, upregulated the expression of PSCA mRNA compared with the C allele of rs22904008 both in normal gastric and gastric cancer tissue, and the prognosis gets worse with increased mRNA expression. These contradictory results may be due to the qualitative traits change in PSCA. The rs2294008T produces a long PSCA, which has nine amino acids longer than the short PSCA produced by the rs2294008C. The long PSCA is located on the membrane and has a growth-promoting ability, whereas the short PSCA is in the cytoplasm and is easy to be disassembled. The signal analysis of long PSCA is not fully elucidated. Further investigation is required.”
“GWAS analysis reveals that nine PSCA SNPs, including rs2294008 and rs2920283, are identified as risk SNPs for gastric atrophy (GA) and severe GA [34]. Sophie K.M. Hein-richs et al. had conducted the eQTL analysis using 143 subjects of German descent with intestinal metaplasia [35]. They reported that the lead eQTL for PSCA is rs2920283, and an upregulated PSCA mRNA expression in gastric tissue was observed in gastric cancer risk allele carriers. Although we don’t have data indicating the frequency of genotype at rs2920283, rs2920280, which has linkage distribution (D’=0.971, r2= 0.600) to the rs2920283 according to the Ensembl (https://asia.ensembl.org/Homo_sapiens/Info/Index), may in-crease PSCA mRNA. However, the localization and function of PSCA with the rs2920280G are not fully elucidated. Further experiments are needed.”
Reviewer 3 Report
Line 77: I am also wondering why the study is referring to individuals enrolled in 2013. When was the study completed. What is the end date if enrollment was in 2013.
Line 134: please specify the three inheritance models in writing before referencing table 2. The three models are clear in the table, but it is better to specify them in the text as well on line 134.
Line 135: change "between" into "for the"
Line 160: change "between" into "for the"
Please make unify the style for writing P-value all over the manuscript (for example in tables 4 and 5, one P is in italics and the other is not. Place dash between the p and value (i.e., P-vale)
The authors need to expand on the discussion section:
- On line 210, the authors mention that this SNP may be a useful prognostic biomarker for GMA….. --> it is advised that the authors discuss what other prognostic biomarkers can be used for the same condition? And what is the advantage of using this SNP above other biomarkers.
- Line 215 --> explain what do you mean by lower sensitivity? Lower sensitivity of what? Write down this clearly.
- Provide more details on intron 1? Specify the gene (intron 1 of what gene?), or/and the genomic coordinates.
- Line 231: "within the same gene within a short distance " --> fix and clarify this sentence. Specify the gene "were located in the same gene (i.e., name the gene) within a short distance from each other,
- Line 231: Reword this sentence: "and it could not be denied completely being in linkage disequilibrium. "
The authors can discuss if combining this biomarker with other biomarkers (such as imaging biomarkers, Machine learning, microbiome, etc.) can be valuable to patients? Explain the value of using these SNPs with other biomarkers in these patients? You can benefit from the following references (and others of your preference) to identify biomarkers that could be combined with this SNP, and indicate the advantages of such combinations.
https://doi.org/10.3390/jcm11123523
https://www.mdpi.com/2075-4418/12/5/1214
https://www.mdpi.com/2075-4418/12/1/133/htm
https://www.mdpi.com/2075-4418/11/5/742
Author Response
Line 77: I am also wondering why the study is referring to individuals enrolled in 2013. When was the study completed. What is the end date if enrollment was in 2013.
Thank you for your question. The end day of enrollment is December 10 in 2013. In 2013, the Japanese insurance for the first time covered eradication therapy of H. Pylori even for chronic gastritis alone without any significant disorders such as peptic ulcers and lymphomas. Therefore, we conducted this study in 2013 to enroll individuals with chronic gastritis alone.
Line 134: please specify the three inheritance models in writing before referencing table 2. The three models are clear in the table, but it is better to specify them in the text as well on line 134.
Thank you for your advice. I changed the sentences as follows.
“The frequencies of alleles and genotypes of the five tag SNPs in PSCA showed no significant differences in the allele and genotype frequencies between the GMA (n = 92) and non-GMA (n = 103) groups in the three inheritance models; allele model, dominant model, and recessive model (Table 2).”
Line 135: change "between" into "for the"
I changed “between” into “for the”.
Line 160: change "between" into "for the"
changed “between” into “for the”.
Please make unify the style for writing P-value all over the manuscript (for example in tables 4 and 5, one P is in italics and the other is not. Place dash between the p and value (i.e., P-vale)
I appreciate. I change as P-value.
The authors need to expand on the discussion section:
- On line 210, the authors mention that this SNP may be a useful prognostic biomarker for GMA….. --> it is advised that the authors discuss what other prognostic biomarkers can be used for the same condition? And what is the advantage of using this SNP above other biomarkers.
I changed the sentence as follows.
SNP rs2920280 may be useful as a prognostic biomarker for detecting individuals at-risk of developing GMA among H. pylori-infected individuals than SNP rs2294008.
- Line 215 --> explain what do you mean by lower sensitivity? Lower sensitivity of what? Write down this clearly.
I changed this sentence as follows.
The lower sensitivity than the biomarker of rs2294008 implies the detection of only one-quarter of individuals developing GMA.
- Provide more details on intron 1? Specify the gene (intron 1 of what gene?), or/and the genomic coordinates.
I changed as follows.
“SNP rs2920280 is located within intron 1 of PSCA (Figure 1).”
- Line 231: "within the same gene within a short distance " --> fix and clarify this sentence. Specify the gene "were located in the same gene (i.e., name the gene) within a short distance from each other.
I change the sentence as follows.
“We could not deny that rs2294008 and rs2920280 are completely in linkage disequilibrium since these two SNPs were located within a short distance from each other.”
- Line 231: Reword this sentence: "and it could not be denied completely being in linkage disequilibrium. "
I changed the sentence as follows.
“We could not deny that rs2294008 and rs2920280 are completely in linkage disequilibrium since these two SNPs were located within a short distance from each other.”
The authors can discuss if combining this biomarker with other biomarkers (such as imaging biomarkers, Machine learning, microbiome, etc.) can be valuable to patients? Explain the value of using these SNPs with other biomarkers in these patients? You can benefit from the following references (and others of your preference) to identify biomarkers that could be combined with this SNP, and indicate the advantages of such combinations.
https://doi.org/10.3390/jcm11123523
https://www.mdpi.com/2075-4418/12/5/1214
https://www.mdpi.com/2075-4418/12/1/133/htm
https://www.mdpi.com/2075-4418/11/5/742
Thank you for your advice. I added the following sentences.
Although early detection and eradication of H. Pylori infection in adolescents is important, the benefit and harm of the eradication therapy should be always considered. And more, these two SNPs biomarkers would be useful if combined with the ABC methods, which are used to evaluate conditions of the stomach based on the noninvasive methods of measuring pepsinogen and H. Pylori antibody. Stratifying the patients between the high-risk group and the low-risk group would be helpful for deciding whether the patients should take medicine or encourage frequent endoscopic surveillance for gastric cancer identification.
Round 2
Reviewer 1 Report
The author's response to the reviewer's comments was adequate, and the revised manuscript was appropriately refined. As a result, this report is of great interest.Reviewer 2 Report
The revised manuscript has greatly been improved and it is now acceptable in the journal.